# Towards the Application of Purely Inorganic Icosahedral Boron Clusters in Emerging Nanomedicine

**DOI:** 10.3390/molecules28114449

**Published:** 2023-05-30

**Authors:** Francesc Teixidor, Rosario Núñez, Clara Viñas

**Affiliations:** Institut de Ciència de Materials de Barcelona, ICMAB-CSIC, 08193 Bellaterra, Spain; teixidor@icmab.es

**Keywords:** carboranes, metallabis(dicarbollide), BNCT, proton therapy, PBFR, COSAN, FESAN, PET, SPECT, antimicrobial, luminescence, bioimaging, photodinamic therapy (PDT)

## Abstract

Traditionally, drugs were obtained by extraction from medicinal plants, but more recently also by organic synthesis. Today, medicinal chemistry continues to focus on organic compounds and the majority of commercially available drugs are organic molecules, which can incorporate nitrogen, oxygen, and halogens, as well as carbon and hydrogen. Aromatic organic compounds that play important roles in biochemistry find numerous applications ranging from drug delivery to nanotechnology or biomarkers. We achieved a major accomplishment by demonstrating experimentally/theoretically that boranes, carboranes, as well as metallabis(dicarbollides), exhibit global 3D aromaticity. Based on the stability–aromaticity relationship, as well as on the progress made in the synthesis of derivatized clusters, we have opened up new applications of boron icosahedral clusters as key components in the field of novel healthcare materials. In this brief review, we present the results obtained at the Laboratory of Inorganic Materials and Catalysis (LMI) of the Institut de Ciència de Materials de Barcelona (ICMAB-CSIC) with icosahedral boron clusters. These 3D geometric shape clusters, the semi-metallic nature of boron and the presence of *exo*-cluster hydrogen atoms that can interact with biomolecules through non-covalent hydrogen and dihydrogen bonds, play a key role in endowing these compounds with unique properties in largely unexplored (bio)materials.

## 1. Introduction

Boron was isolated in Penzance (Cornwall, England) in 1808 by the English chemist Humphry Davy [1], but boron as an element was identified by Jöns Jakob Berzelius in 1824 [2]. Boron is extracted as borate salts of different cations from minerals (Kaliborite, Karlite, Kernita and Kurnakovite, among others) [3,4]. Turkey (deposits existing in Kırka, Emet, Bigadiç, and Kestelek) has the largest world boron reserves, followed by the United States (“Death Valley” desert in California) and Russia at the second position [5], being Turkey the major country in boron production from 2010 to 2022 [6].

Borax (Na_2_B_4_O_7·_10H_2_O) was one of the first minerals to be exchanged in the times of the Ancient World. In the Egypt of the phaaraohs, the deceased were embalmed with mummification salts, being those containing borate the most reliable for preservation. Boric acid (H_3_BO_3_), which was produced from borax by the Dutch chemist William Homberg in 1702, has been widely used for topical administration since the 18th century due to its strong bactericidal and fungicidal activity [7].

Boron, which is located to the left of carbon on the periodic table, possesses and forms stable compounds with a wide variety of elements. Natural boron is composed of two stable isotopes ^10^B and ^11^B, the latter of which make up about 80% of natural boron. Boron, like carbon, can bond with itself, forming B-B bonds that give rise to boranes and heteroboranes (being the most known carboranes and metallacarboranes). These boron clusters form 3D aromatic [8,9,10], polyhedral structures with triangular faces in which the bonds that hold the cluster together are tricentric bonds with two electrons (3c-2e). William Lipscomb received the Nobel Prize in Chemistry in 1976 for his studies on the 3c-2e bonding of borane structures [11]. These 3D molecular structures of boron clusters possess extraordinary chemical, biological, thermal, and photochemical stability that make them have unique applications in (nano)materials not possible with other elements, including carbon [12,13,14,15,16,17].

The traditional use of organic chemistry as the basis for all aspects of contemporary biomedical chemistry has provided truly miraculous results. Nowadays, most commercial drugs are purely organic molecules, but nitrogen, oxygen, phosphorus, sulfur, and halogens, all neighbors of carbon to the right, are part of a wide variety of the active principles of medicines. In the middle of the 20th century [18,19,20,21], the first investigations of boron compounds for their use in medicine were directed mainly towards the treatment of cancer by the therapy called BNCT (Boron Neutron Caption Therapy), but currently, a vibrant and growing research is being developed to employ boron-containing compounds in medicinal chemistry and chemical biology [22,23,24,25,26,27,28,29,30,31,32,33].

This mini-review focuses on the large research activity of the Inorganic Materials and Catalysis Laboratory (LMI) at the Institut de Ciència de Materials de Barcelona (ICMAB-CSIC) [34] with icosahedral boron clusters, which due to their geometric shape and the semi-metal nature of boron provide these compounds with unique properties in (bio)materials largely unexplored.

## 2. Characteristics of Icosahedral Neutral Carboranes and Anionic Metallabis(Dicarbollides)

### 2.1. Icosahedral Closo-Borane and Heteroborane Clusters

Figure 1 shows the inorganic icosahedral *closo-*dodecaborate ([B_10_H_12_]^2−^), the dicarba-*closo*-dodecaborane (*closo* C_2_B_10_H_12_), which exists in three isomeric forms that are named based on the positioning of the two CH vertices: 1,2- or *ortho*-, 1,7- or *meta*-, and 1,12- or *para*-carborane and, the sandwich metallabis(dicarbollides) [M(C_2_B_9_H_11_)_2_]^−^ (M = Co^3+^, Fe^3+^). Five different conformations can be found in the metallabis(dicarbollides): *cisoid*-1, *gauche*-1, *transoid*, *gauche*-2 and *cisoid*-2. However, *cisoid*-1 and *cisoid*-2, as well as *gauche*-1 and *gauche*-2, are equivalent in the non-substituted or symmetrically disubstituted clusters [35].

Teixidor and Viñas believed that one of the main reasons for the lack of knowledge and poor application of these boron compounds is the lack of synthetic processes for their functionalization. Without these processes, the chemistry of boron is marginalized when its possibilities are enormous, and in many cases, complementary to the organic chemistry compounds.

### 2.2. Towards the Derivatization of the Icosahedral Boron Clusters

The neutral icosahedral *closo* C_2_B_10_H_12_ carboranes have the potential for the incorporation of a large number of substituents at its 12 vertices (2 C-H and 10 B-H). The reactivity of the B-H vertices depends on the distance of each B-H vertex to the C-H ones. Most reactions that occur at the boron vertices do not affect the carbon vertices, and vice versa. Consequently, *o*-carborane offers the possibility to develop chemistry of neutral *closo*-carboranes at the C vertices, at the B vertices, as well as in both C and B vertices (Figure 2) [35].

Since 1982, Teixidor and Viñas have put emphasis on improving protocols of syntheses because their main objective was the application of icosahedral boron clusters [36], and clusters’ derivatization was a necessary and key step to proceed on their use in (bio)materials [37]. Recently, several reviews summarizing the different synthetic procedures to achieve the substitution at the cluster vertices of the icosahedral boron clusters appeared [38,39,40,41,42,43,44,45,46,47].

## 3. Focusing on the Synthesis of Icosahedral Neutral Carborane and Anionic Metallabis(Dicarbollide) Derivatives for Medicinal Application

Teixidor and Viñas group carried out remarkable work in the synthesis of icosahedral carborane and metallacarborane derivatives as well as in their characterization with the objective of finding their application in different fields.

Endo and co-workers, based on the similarities between the phenyl group and the carborane cluster, pioneered the design of new drugs by substituting phenyl groups in compounds with known biological activity with icosahedral carborane groups [48,49,50,51,52,53]. The concept of 3D aromaticity has already been applied in boron cluster chemistry to relate the limited number of valence electrons in the clusters to their stability [54,55]. Recently [8,9,10], the 3D global aromaticity of the icosahedral boranes, carboranes, and cobaltabis(dicarbollides) was related to the more familiar 2D aromaticity abiding by Hückel’s rule, indicating that both were two sides of the same coin. Then, in 2014 [7], grounded on the relationship between stability and aromaticity, new perspectives for applying icosahedral boron clusters as key components in the field of new biomaterials for healthcare were opened by Teixidor and Viñas group. The highlight is the development of potentiometric sensors for the detection of drugs [56,57,58,59], biosensors [60], and X-ray contrast agents for highly radiopaque vertebroplasty cement [61], among others.

Special emphasis is given to fostering advances in the application of boron compounds for the Boron Neutron Capture Therapy (BNCT) treatment of cancer due to the inherent property of the boron element itself (with 20% of ^10^B). ^10^B has a large neutron capture section opening up the application of icosahedral boron clusters to the treatment of cancer by the BNCT reaction between a thermal neutron and ^10^B resulting in the generation of an α particle and 7Li nucleus (Figure 1). Additionally, the 3D aromatic icosahedral boron clusters offer the possibility of holding twelve substituents covering the entire 3D space.

The twelve vertices of the cluster can be functionalized, and these small molecules can be converted into multifunctional scaffolds by themselves (Figure 3) [62,63,64,65] or bonded to inhibitors of kinases receptor molecules (Figure 4) [66,67,68,69] or in anchoring onto structures of nanocarriers (dendrimers [28,70], polymers [14], nanoparticles [36,71,72,73,74,75]) leading to payloads with high boron density (Figure 3). The objective was to synthesize anionic and water-soluble high-boron-containing molecules, which can incorporate in their scaffold either inhibitors of enzymes receptor (Figure 4) [66,67,68,69] and/or metal cores (Figure 5) for their use as multifunctional nanocarriers able to act as anticancer drugs by multi-therapy treatment [73,74,75,76,77].

## 4. Testing the Icosahedral Neutral Carboranes and Anionic Metallabis(Dicarbollides) in BNCT Cancer Treatment

### 4.1. Boron Neutron Capture Therapy

Regarding carboranes for BNCT, the research has focused on the development of new multifunctional hybrid (carboranyl + anilinoquinazolines) nanocarriers [66,67] and carborane-magnetic nanoparticles [73]. These (bio)materials exhibit desirable in vitro antitumor activities against preclinical rat glioblastoma F98, colorectal HT29, glioblastoma A172 cancer cell lines, and human brain endothelial hCMEC/D3 cell line.

Importantly, thermal neutrons irradiation in BNCT for 15 min reduced by 2.5 the number of cultured A172 glioblastoma cells after the treatment with carborane-magnetic nanoparticles (Figure 6a) and the systemic administration of carborane-magnetic nanoparticles in mice was well tolerated with no major signs of toxicity. The dual treatment by combining tyrosine kinase inhibition and BNCT irradiation for minutes on HT-29 cells after incubation with carboranyl + anilinoquinazoline hybrids provided better outcomes than *p*-Boronophenylalanine (BPA) [68]. The attractive profile of developed hybrids makes them interesting agents for combined therapy (Figure 6b).

### 4.2. Metallabis(Dicarbollides) Chemical and Physico-Chemical Properties and Cytotoxicity

Regarding the icosahedral metallacarboranes, the anionic metallabis(dicarbollides), [3,3-M(1,2-C_2_B_9_H_11_)_2_]^−^, (abbreviated as [*o*-COSAN]^−^ and [*o*-FESAN]^−^ for M = Co, Fe, respectively), which are inert to biochemical reactions, have attracted much attention in biology [35]. The 3D aromatic Na[*o*-COSAN] forms hydrogen and dihydrogen bonds that participate in its self-assembling, water solubility, and aggregates’ formation [76]. The Na[o-COSAN] possesses the ability to readily cross cell membranes (Figure 7a) [77,78,79], is not cytotoxic against mammalian cells (HEK 293, HeLa, THP-1, 3T3), *D. discoideum* amoeba cells, and bacteria (*E. coli* and *Klebsiella*), but is cytostatic, and cells recover following its removal [79]. Furthermore, our studies on glioma-initiating cells (GIC7 and PG88) also supported Na[*o*-COSAN] cytostatic properties when cells were morphologically recovered 43 h after washing off the compound and increasing in the G2/M subpopulation. Additionally, the study showed that mesenchymal PG88 cells that are more resistant than proneural GIC7 cells to conventional radiotherapy have a lower EC_50_ Na[*o*-COSAN] and a higher uptake of the compound compared to GIC7 cells, suggesting a new resource to fight against resistant glioblastoma cells [80].

### 4.3. Synchrotron-Based Fourier-Transform Infrared Micro-Spectroscopy (SR-FTIRM) Studies

Having performed experiments in a round-bottom flask on a chemical scale, which showed that [*o*-COSAN]^−^ and some of its halogenated derivatives interact with biomolecules (amino acids [56,57], proteins [81,82], *ds*-DNA [60,83] (Figure 7b) and glucose [84]), we wanted to go a step ahead by observing these interactions in vitro experiments by using SR-FTIRM. The round-bottom flask changed to a cell, and the solutions to the cell’s physiological components. The first chemical scale studies between [*o*-COSAN]^−^ anions and the biomolecules were done individually for each type, whereas the cell study incorporates the effect of all biomolecules interacting simultaneously. This study meant a step ahead to understand and detect that this anion modifies biomolecules (proteins, DNA, and lipids) and concentrates in the cell nucleus after their cellular uptake [68]. The small Na[*o*-COSAN] molecule, localized close to the cell’s nucleus, induces proteins’ conformational changes and spectral changes of the DNA region (Figure 7b) in both GIC cell lines, similar to the changes induced by other metal-based compounds like cisplatin that disrupt the double helix base pairing, suggesting that Na[*o*-COSAN] is a promising agent for BNCT of glioblastoma.

Consequently, in vitro tests in U87 and T98G cells conclude that the amount of ^10^B inside the cells is enough for BNCT irradiation. BNCT becomes more effective on T98G after their incubation with Na[8,8′-I_2_-*o*-COSAN], whereas no apparent cell-killing effect was observed for untreated cells.

All this led to the following conclusions: These small molecules, particularly [8,8′-I_2_-*o*-COSAN]^−^, are serious candidates for BNCT now that the facilities of accelerator-based neutron sources are more accessible, providing an alternative treatment for resistant glioblastoma (Figure 7c) [85].

Then, in vivo experiments with Na[*o*-COSAN] and Na[8,8′-I_2_-*o*-COSAN] were performed on *Caenorhabditis elegans (C. elegans)* at the L4-stage and their embryos. LD50 values for both cobaltabis(dicarbollides) in L4 *C. elegans* were found to be close to the IC_50_ determined for T98G in vitro after 72 h (Figure 8) [86].

Finally, in vivo evaluation in mammalian mice models were run trying to understand the ability of [*o*-COSAN]^−^ to target the tumor cells, as well as to cross the blood–brain barrier. After intravenous administration, biodistribution studies of Na[*o*-COSAN] in BALB/c CrSlc mice (female, 5 weeks old) were run. Anionic [*o*-COSAN]^−^ was distributed into many organs but mainly accumulated in the reticuloendothelial system (RES), including liver and spleen (Figure 9) [83].

### 4.4. Contrast Agents

Furthermore, Na[8-I-*o*-COSAN] can be labeled with contrast agents, such as ^124^I and ^125^I, for in vivo markers by positron emission tomography (PET) and single photon emission computed tomography (SPECT) nuclear imaging techniques making these clusters very good scaffolds as theranostic agents (Figure 10) [87].

The synthesis of these unprecedented radiolabeling Na[8-I-*o*-COSAN] anionic derivatives with either ^125^I (gamma emitter) or ^124^I (positron emitter) was achieved via palladium-catalyzed isotopic exchange reaction (Figure 2a) following our previously reported synthesis of ^125^I carborane derivatives (2-I-*p*-, 3-I-*o*-, 9-I-*o*-, 9-I-*m*-carborane, 1-phenyl-3-I-*o*-carborane, and 1,2-diphenyl-3-I-*o*-carborane) with some modifications (Figure 2b) [87].

Recently, the sodium salt of the anionic [*o*-FESAN]^−^ isotopically 100% ^57^Fe was synthesized with the objective of treating glioblastoma cancer with Na[3,3′-^57^Fe(1,2-C_2_B_9_H_11_)_2_] because the compound offers the possibility of dual-action (radiation + drug combinations) to improve clinical benefits and reduce healthy tissues toxicity. After [*o*-^57^FESAN]^−^ uptake by U87 glioblastoma cells, [*o*-^57^FESAN]^−^ was found to be within the cells with 29% of its uptake in the nuclear fraction, which is a particularly desirable target because the nucleus is the cell control center in which DNA and transcription machinery reside. The multi-therapies activity through irradiation with potential for glioblastoma treatment by the Mossbauer effect of [3,3′-^57^Fe(1,2-C_2_B_9_H_11_)_2_]^−^ was demonstrated (Figure 11) [88].

### 4.5. Proton Therapy Based on Boron

Proton therapy is an effective radiation treatment technique used in medicine, which consists of irradiating diseased tissue, most often to treat cancer, with a beam of protons [89,90]. Figure 3 represents the Proton Boron Fusion Reaction (PBFR) between an energetic proton and ^11^B resulting in the generation of three α particles. The viability of applying the proton boron fusion (PBF) reaction to the proton therapy to improve its effectiveness has been studied by using the Monte Carlo method [91,92,93] and experimentally using mercaptoundecahydro-closo-dodecaborate (abbreviated as BSH, which chemical formula is [SH-1-*closo*-B_12_H_11_]^−^) [94]. Recently [95], taking advantage of the high ^11^B isotope content in metallabis(dicarbollides), we tested, for the first time, metallacarboranes for the PBFR as a way to improve proton therapy with the [*o*-FESAN]^−^ in the U87 glioblastoma cells. A simple calculation indicates that the use of PBFR would require 1/12 of isotopically natural molecules with respect to BNCT. Furthermore, in an ideal situation, BNCT can be used synchronously on the existing ^10^B and Mössbauer on ^57^Fe, resulting in multi therapies with only one compound. Results from the cellular damage response obtained suggest that PBFR radiation therapy, when applied to boron-rich compounds, is a promising modality to fight against resistant tumors.

### 4.6. Antimicrobial Activity

In 2013, we started studying the physical–chemical properties and biological evaluation of the sodium salt of the small inorganic metallabis(dicarbollide) molecules ([*o*-COSAN]^−^ and [*o*-FESAN]^−^) and their derivatives [8-R(CH_2_CH_2_O)_2_-*o*-COSAN)]^−^ (R = -OOCCH_3_; -OCH_3_; -OCH_2_CH_3_) against pure cultures of 16 pathogenic bacterial strains (isolated from animals and humans as well as control strains) and 3 strains of *Candida* spp. as promising antimicrobial agents to tackle bacterial infections [96]. It is important to emphasize that the methicillin-resistant strain of *Staphylococcus aureus* (MRSA), the polyresistant strains of *Pseudomonas aeruginosa*, as well as of *Candida* spp., are sensitive to the compounds Na[8-CH_3_(CH_2_CH_2_O)_2_-*o*-COSAN)] and Na[8-CH_3_CH_2_(CH_2_CH_2_O)_2_-*o*-COSAN)]. Recently, a review of the increasing evidence that boron cluster compounds are promising antimicrobial (antibacterial and antifungal) agents appeared [97]. Lately, with the objective to establish a structure–activity relationship, which clearly supports the antimicrobial activity of the pristine metallabis(dicarbollide) complexes, we tested the small molecules Na[*o*-COSAN], Na[*o*-FESAN], Na[*m*-COSAN)], Na[*m*-FESAN)], the di-iodinated derivatives Na[8,8′-I_2_-*o*-COSAN], Na[8,8′-I_2_-*o*-FESAN] and polyanionic species incorporating one or two cobaltabis(dicarbollide) anions with activity against four *Gram-positive* bacteria (two *Enterococcus faecalis* strains and two of *Staphylococcus aureus* including Multi-Resistant Staphylococcus Aureus (MRSA) strains), five *Gram-negative* bacteria (three strains of *Escherichia coli* and two of *Pseudomonas aeruginosa*), and three *Candida albicans* strains that have been responsible for human infections [98,99]. We demonstrated an antimicrobial effect against *Candida* species (Minimum Inhibitory Concentration (MIC) of 2 and 3 nM for Na[8,8′-I_2_-*o*-COSAN] and Na[*m*-COSAN], respectively), and against *Gram-positive* and *Gram-negative* bacteria, including multi-resistant MRSA strains (MIC of 6 nM for Na[8,8′-I_2_-*o*-COSAN]). The selectivity index (abbreviated as SI and, calculated as the ratio IC_50_/MIC) for antimicrobial activity of Na[*o*-COSAN] and Na[8,8′-I_2_-*o*-COSAN] compounds is very high (165 and 1180, respectively), which reveals that these small anionic metallacarborane molecules may be useful to tackle antibiotic-resistant bacteria because it is considered that an SI ≥ 10 is acceptable for a selective bioactive sample.

Furthermore, we demonstrated that the outer membrane of *Gram-negative* bacteria establishes an impermeable barrier for some of these metallabis(dicarbollide) small molecules (Figure 4). Nonetheless, the addition of two iodine groups in the structure of the parent Na[*o*-COSAN] had an improved effect (3–7 times) against *Gram-negative* bacteria. It is important to emphasize that the most active metallabis(dicarbollides) (*meta*-isomers Na[*m*-COSAN)], Na[*m*-FESAN)] and the di-iodinated derivatives Na[8,8′-I_2_-*o*-COSAN], Na[8,8′-I_2_-*o*-FESAN]) are both *transoid* conformers in opposite to the Na[*o*-COSAN] that is *cisoid* conformer (see Figure 1), which represent structures with particular physical–chemical properties that make these small molecules more permeable to this barrier.

The fact that these small molecules cross the mammalian membrane and have antimicrobial properties but low toxicity for mammalian cells (high selectivity index SI) represents a promising tool to treat infectious intracellular bacteria as there is an urgent need for new antibiotics discovery and development. This achievement represents a relevant advance in the field.

## 5. Boron Clusters-Based Dyes as Theranostic Agents for Diagnosis and Therapy

Today, one of the most important tools in predicting disease is diagnosis. Molecular imaging is a remarkable diagnostic tool in vitro and in vivo that could provide crucial biological information regarding a targeted disease and can thus help to establish a particular treatment or therapy [100,101]. Moreover, the development of theranostic systems to integrate imaging and therapy is an efficient strategy for real-time tracking of the pharmacokinetics and biodistribution of a drug. Current imaging modalities include optics (e.g., fluorescence, Raman, photoacoustics), X-ray, magnetic resonance, radionuclides, and mass spectrometry [102]. Among them, Fluorescence Bioimaging is a common modality for cell and tissue visualization, being of special interest in preclinical research on theranostic agents. In this context, each fluorophore has its benefits and drawbacks, which requires the continued search for new fluorescent probes to meet stringent necessities for applications in terms of sensitive and selective use for bioimaging applications.

Moreover, imaging-guided BNCT is a challenge as it allows us to know the accurate position of the boron-containing compound in the body as well as the accumulation in the tumor. Therefore, it is an important issue to label the boron-containing compound with a fluorescence tracer in order to have relevant information for both diagnosis and therapy [103,104,105,106,107]. In particular, the near-infrared (NIR) boron carriers are of great interest due to their deeper penetration into the living body and their ability to avoid interference from body tissues [108,109].

Dr. Núñez has been a staff member of the LMI group since 2001. Over this time, she has developed synthetic strategies for the functionalization of a great variety of scaffolds, i.e., star-shape molecules and dendrimers [28,110], octasilsesquioxanes [111,112,113,114], carbon-based materials [115,116], among others [117], with icosahedral boron clusters and studied their properties. In 2007, Dr. Núñez reported a set of blue emissive Fréchet-type aryl ether core molecules peripherally functionalized with *closo*-carborane and *nido*-carborane clusters [118,119]. It was then demonstrated that the maximum wavelength and emission intensity depend on the C_cluster_ substituent (Me or Ph), the solvent polarity, and the nature of the cluster (*closo* or *nido*). This work was the beginning of her immersion in the field of luminescence. Since then, her main interest has been the development of photoluminescent boron cluster-based organic π-conjugated dyes [120,121,122,123,124,125,126,127,128], revealing that incorporation of neutral and anionic boron clusters into the structure of well-known fluorophores is an attractive chemical strategy to modulate and improve their photoluminescence properties [15,120,121,122,123,124,125,126,127,128]. Her current research interest is more focused on new boron-based molecules and materials as theranostic agents for diagnosis (bioimaging) and boron carriers for BNCT.

A set of BODIPY-anionic boron cluster conjugates bearing dianionic [B_12_H_12_]^2−^ and monoanionic, [*o*-COSAN]^−^ and [*o*-FESAN]^−^ clusters were designed and synthesized to be used as fluorescent cell probes and BNCT anticancer agents (**1**–**5** in Figure 12a) [129]. These conjugates were readily synthesized from the meso-(4-hydroxyphenyl)-4,4-difluoro-4-bora-3a,4a-diaza-s-indacene (BODIPY) by ring-opening reaction of the corresponding boron clusters derivatives. The luminescent properties of the BODIPY were not significantly altered by the linking of the anionic boron clusters, showing emission fluorescent quantum yields (Φ_F_) in the range of 3–6%. Moreover, the cytotoxicity and cellular uptake of these compounds were analyzed in vitro at different concentrations of B (5, 50, and 100 µg B/mL) using HeLa cells. None of the compounds showed cytotoxicity at the lowest concentration (5 µg B/mL). Compound bearing [B_12_H_12_]^2−^ and Na^+^ as cation were non cytotoxic at any concentration, while the other compounds showed toxicity at the highest concentrations after 24h. Remarkably, all the compounds were successfully internalized by HeLa cells, exhibiting a strong cytoplasmic stain (Figure 12b). The internalization efficiency for all the compounds was assessed at the lowest concentration (5 µg B/mL), in which they are not cytotoxic. The exceptional cellular uptake and intracellular boron release, together with their fluorescent and biocompatibility properties, highlight the suitability of these boron cluster-containing dyes, especially [*o*-COSAN]^−^ derivative, as potential candidates for cell labeling agents towards medical diagnosis in clinical biopsies. Moreover, the excellent cellular uptake, along with the boron-rich content of our conjugates, make them good candidates as boron carriers for BNCT.

Our interest in the development of new boron delivery systems to be used for biological applications led us to prepare a family of fluorescent organotin compounds that have shown excellent properties as nucleoli and cytoplasmic markers in vitro [130]. These organotin compounds are based on 4-hydroxy-N′-((2-hydroxynaphthalen-1-yl)methylene)benzohydrazidato that was derivatized to contain two different boron clusters, [B_12_H_12_]^2−^ and [*o*-COSAN]^−^ following the oxonium ring opening reaction (**6**–**9** in Figure 13a). These compounds showed photoluminescence properties in solution with Φ_F_ values in the range from 24% to 49%. Remarkably, linking these anionic boron clusters to tin complexes improved their solubility in cell media, which resulted in better cell internalization and higher cellular uptake, as they do not aggregate either on the cell surface or in the extracellular media. Mouse melanoma B16F10 cells were incubated with 10 µg/mL of the different compounds for 2 h and then analyzed by confocal laser microscopy. Noticeably different staining effect was observed depending on the type of boron cluster attached to the organotin complexes. Compound **8** bearing the [*o*-COSAN]^−^ anion and two phenyl rings coordinated to the Sn showed an important fluorescence in the cytoplasm, whereas that bearing [B_12_H_12_]^2−^ (**6**) produced extraordinary nucleoli and cytoplasmic staining (Figure 13b). The remarkable fluorescence staining properties of these organotin compounds in B16F10 cells make them excellent candidates for in vitro fluorescent bioimaging.

Apart from previous BODIPY derivatives bearing anionic boron clusters, our group has also developed a family of neutral BODIPY-carboranyl conjugates which have been synthesized following Sonogashira or Heck cross-coupling reactions in which properly functionalized *ortho*- and *meta*-carborane clusters have been linked to light-emitting BODIPY or aza-BODIPY cores [131,132,133]. Figure 14 illustrates three different BODIPY-carboranyl systems with Ph-*ortho*-carborane (**10**–**11**) and Ph-*meta*-carborane (**12**), as examples. Due to their fluorescence properties, these fluorophores were studied in vitro as fluorescent probes. HeLa cells were incubated for 30 min with this set of BODIPYs, which presented very different behavior regarding cellular uptake and subcellular distribution (Figure 14) [132]. The differences seem to originate from their diverse static dipole moments and partition coefficients, which depend on the type of cluster isomer (*o*- or *m*-) linked to the BODIPY and that modulates the ability of these molecules to interact with the lipophilic microenvironments in cells. It can be highlighted that the *m*-carborane derivative with higher lipophilicity was much better internalized by cells than their *ortho* analogs. Confocal images of HeLa cells incubated with **12** (Figure 14) clearly indicate that **12** is accumulated in the cytoplasm of the cell. This evidence provides a molecular design strategy for improving the prospective applications of BODIPY-carboranyl dyads as potential fluorescence in vitro bioimaging agents and boron carriers for BNCT, suggesting that *m*-isomers are potentially better theranostic agents than *o*-isomers.

Another type of well-known fluorophores are anthracene derivatives that exhibit excellent luminescence properties that make them perfect scaffolds for optical applications. Our group has developed efficient blue light-emitting materials by combining the properties of anthracene and *m*-carborane [134]. Three different *m*-carborane-anthracene dyads, in which the carborane is non-iodinated, mono-iodinated, or di-iodinated at B atoms, and the anthracene fragment is linked to one C_cluster_ atom through a CH_2_ spacer, were prepared. All of them exhibited exceptional fluorescence properties with high quantum yields (Φ_F_ ~ 100%) in solution with maximum emission of around 415 nm, confirming that simply linking the *m*-carborane fragment to one fluorophore produces a significant enhancement of the fluorescence emission in the target compound. Notably, the three conjugates exhibited good fluorescence efficiencies in aggregate state with Φ_F_ in the range 19–23%, indicating that our dyads are extremely good emitters in solution, while maintaining the emission properties in the aggregate state. Moreover, their cytotoxicity and cellular uptake in HeLa cells were evaluated. None of the compounds showed cytotoxicity at different concentrations for HeLa cells. Confocal microscopy studies confirmed that, although all compounds were internalized by cells via endocytosis, exhibiting high fluorescence emission intensity, the one with two iodo atoms is the one with a higher cellular uptake. This suggested that the presence of iodo units leads to a more efficient transport across the plasma membrane and a better internalization of the compounds. Figure 15 shows the autofluorescence of HeLa cells and fluorescence emitted by Hela cells incubated with the diiodinated antracece-*m*-carborane. We then conclude that the di-iodinated compound is an excellent candidate as a fluorescent dye for bioimaging studies in fixed cells, and due to the high boron content and exceptional cellular uptake, it could be used as a potential anticancer agent for BNCT.

Besides previous luminescent materials, our group has also prepared carbon-based nanomaterials, which consist of graphene oxide (GO) functionalized on the surface by monoiodinated cobaltabis(dicarbollide) (GO-I-COSAN) for in vivo bioimaging. This GO-I-COSAN has been synthesized using the cobaltabis(dicarbollide) containing a B-I group and an amino group (I-COSAN) that is subsequently labeled with radioactive ^124^I (Figure 5) for its use in positron emission tomography (PET) [135]. After incubation of HeLa cells with different concentrations of GO-I-COSAN for 48 h, the results indicated that the nanomaterial was not cytotoxic, with cell mortality lower than 10%. Remarkably, internalization of the nanomaterial by cells was clearly confirmed by transmission electron microscopy (TEM), which showed that the GO-I-COSAN was accumulated in the cytoplasm without causing changes in either the size or morphology of the cells. Further in vivo studies using *C. elegans* indicated that GO-I-COSAN was ingested by the worms, showing no significant damage and very low toxicity, which supports the results observed in vitro. Radioisotopic labeling of I-COSAN using a palladium-catalyzed isotopic exchange reaction with Na[^124^I]I and its subsequent functionalization onto GO was performed successfully, leading to the formation of the radioactive nanocomposite GO-[^124^I]I-COSAN (Figure 5). The radiolabeled nanomaterial was injected into the mice, and PET images at different times were taken (Figure 16), which revealed no activity in the thyroid and stomach even at long times, indicating that iodide did not detach from the material. GO-[^124^I]I-COSAN presented a favorable biodistribution profile, with long residence time on blood, mainly accumulated in the liver and slightly in the lung, and progressive elimination via the gastrointestinal tract. It is noteworthy that the high boron content of this material paves the way toward theranostics because it benefits traceable boron delivery for BNCT.

Our group, in collaboration with S. Draper’s group in Dublin, has also reported the preparation of transition metal-carborane photosensitizers by Sonogashira cross-coupling of (4-ethynylbenzyl)methyl-*o*-carborane with halogenated Ru(II)- or Ir(III)-phenanthroline complexes [136]. The resulting carboranyl-containing complexes (RuCB, IrCB, RuCB2, and IrCB2 in Figure 17) exhibited phosphorescence emission with maxima between 630 and 665 nm and lifetimes of 2.53, 0.38, 1.83, and 0.19 μs, respectively. All of them produce singlet oxygen with quantum yields (Φ_Δ_) of 52%, 25%, 20%, and 10%, respectively, which suggests their use as triplet photosensitizers for photodynamic therapy (PDT). The subcellular uptake of all complexes was explored in SKBR-3 cells. Their localization and intensities were different depending on the number of carborane moieties and the nature of the transition metal centers. Complex IrCB was the best internalized with a clear accumulation in the cytoplasm. On the other hand, RuCB was hard to observe in the confocal microscopy images, but further microscopy experiments performed at a higher laser power showed that, in fact, RuCB was internalized. RuCB2 formed aggregates mainly located at the plasma membrane, whereas IrCB2 was poorly detected inside the cell (Figure 18). All of them showed the absence of dark toxicity under photodynamic therapy (PDT) conditions. Despite significant differences in the photophysical activities and cellular internalization of RuCB and IrCB, irradiation (λ_ex_ 405 nm; 3 min; mean intensity 55 µW) of both killed ∼50% of SKBR-3 cells at 10 μM.

## 6. Conclusions

The progress in the synthesis of icosahedral boron clusters and their derivatives, the improvements in particles technology, the advances in medical imaging and computing, and the fact that new irradiation facilities are becoming available at hospitals makes radiotherapies such as BNCT and PBFR viable choices for new cancer medical therapies especially indicated for tumors resistant to chemotherapy and conventional radiotherapy. All this evidence promises to make BNCT and PBFR cutting-edge technology readily more accessible in the near future.

The fact that the icosahedral metallabis(dicarbollide) clusters reported in this review cross the mammalian membrane and have antimicrobial properties but low toxicity for mammalian cells (high selectivity index, SI) represents a promising tool to treat infectious intracellular bacteria. As there is an urgent need for antibiotic discovery and development, these small anionic molecules represent relevant and promising antimicrobial agents to tackle bacterial infections.

This review also gathers several families of boron clusters-based fluorophores with luminescent properties as potential theranostic agents for bioimaging and BNCT. Among them are a series of BODIPYs functionalized with either neutral or anionic boron clusters, a set of anthracene-*m*-carborane dyads, and a family of tin complexes linked to anionic boron clusters. All of them showed excellent fluorescence emission and high cellular uptake. The preparation and study of GO functionalized with radiolabeled cobaltabis(dicarbollide) for PET are described. To end, a set of Ru(II) and Ir(III)-phenanthroline photosensitizers bearing one or two Me-*o*-carborane cages, as well as the in vitro studies for PDT are reported.

## Data Availability

Not applicable.

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
