# Peer review of "Towards the Application of Purely Inorganic Icosahedral Boron Clusters in Emerging Nanomedicine"

_molecules, 2023, doi:10.3390/molecules28114449_

Round 1
Reviewer 1 Report
The manuscript reviews results obtained at the Laboratory of Inorganic Materials and Catalysis (LIMC) of the Institut de Ciència de Materials de Barcelona (ICMAB-CSIC). In particular, the authors focus on the main uses of polyhedral boron-containing compounds in medical treatments such as Boron Neutron Capture Therapy (BNCT), Proton Boron Fusion Reactions (PBFR), the unique membrane translocation properties of some boron clusters, their antimicrobial activity, as well as photophysical activities that make them also useful theranostic agents for bioimaging.
The article is interesting and I recommend its publication in this special issue in Celebration of Professor John D. Kennedy’s 80th Birthday.
However, I would like to make some comments and recommendations:
- Page 1: lines 39, 40, avoid repetition of “the periodic table”; line 42, “Carbon” should be “carbon”; line 44, “These boron clusters are formed by ·3D aromatic, polyhedral structures…” would read better as “These boron cluster form ·3D aromatic, polyhedral structures…”.
- Page 3, lines 109-116: this paragraph is not very clear and I suggest to re-write it, and improve the transition with the preceding paragraph (lines 103 to116), which is too abrupt for the reader.
Although the LIMC group of the ICMAB-CSIC have stablished a connection between the Hückel rule for 2D classical aromaticity and the Wade-Mingos rule for ·3D aromaticity in polyhedral boron-containing compounds, the tridimensional aromaticity of boron clusters has been long recognized and dealt with by other authors. Therefore, the authors of this mini-review should cite previous papers that treat the concept of ·3D aromaticity (and even ‘superaromaticity’), based on the delocalization of the formally insufficient number of skeletal electrons in boranes, carboranes, metallaboranes and metallacarboranes.
- Page 5, figure 4 does not show “New hybrid neutral an anionic boron cluster-containing quinazoline’ molecules…”; figure 4 shows the structure of relevant inhibitors, and individual boron clusters, and it is misleading.
- Page 5, line 161: define “BPA”.
- Page 6, line183: correct “care ytostatic”
- Page 11, line 321: correct “bacteriAs”; line 339: define “LMI”
- Page 15, line 439: define “GO”.
- Page 16, lines 457, 458: the sentence is unclear.
I suggest to organize the manuscript further by adding subsections that should help the readers. For example, the section 4 “Testing the icosahedral…” could be divided by subheadings such as “Boron Neutron Capture Therapy”, “Metallabis(dicarbollides) cytotoxicity”, “SR-FTIRM studies”, “Contrast agents”, “Proton therapy, “antimicrobial activity”. Similarly, the authors could divide section 5 in subsections for clarity.
There are some spelling errors and sentences that are not very clear and the authors could improve.
Author Response
Review 1:
Thanks so much the referee for his/her report and the comments and recommendations that authors appreciate very much.
- Page 1: lines 39, 40, avoid repetition of “the periodic table”; line 42, “Carbon” should be “carbon”; line 44, “These boron clusters are formed by ·3D aromatic, polyhedral structures…” would read better as “These boron cluster form ·3D aromatic, polyhedral structures…”.
The recommendations have been done in the revised file.
- Page 3, lines 109-116: this paragraph is not very clear and I suggest to re-write it, and improve the transition with the preceding paragraph (lines 103 to116), which is too abrupt for the reader.
This paragraph has been amended as referee suggested.
Although the LIMC group of the ICMAB-CSIC have stablished a connection between the Hückel rule for 2D classical aromaticity and the Wade-Mingos rule for ·3D aromaticity in polyhedral boron-containing compounds, the tridimensional aromaticity of boron clusters has been long recognized and dealt with by other authors. Therefore, the authors of this mini-review should cite previous papers that treat the concept of ·3D aromaticity (and even ‘superaromaticity’), based on the delocalization of the formally insufficient number of skeletal electrons in boranes, carboranes, metallaboranes and metallacarboranes.
We wrote the article as a letter to a friend to whom we write to explain our own results during the last years in which Prof. Kennedy has been retired but, the referee is right that we include some references in this point from other researchers. We have done it.
- Page 5, figure 4 does not show “New hybrid neutral an anionic boron cluster-containing quinazoline’ molecules…”; figure 4 shows the structure of relevant inhibitors, and individual boron clusters, and it is misleading.
We have amended this point and the new Figure 4 in the revised file is clearer. Thanks the referee for his/her suggestion.
- Page 5, line 161: define “BPA”.
- Page 6, line183: correct “care ytostatic”
- Page 11, line 321: correct “bacteriAs”; line 339: define “LMI”
- Page 15, line 439: define “GO”.
- Page 16, lines 457, 458: the sentence is unclear.
The mistakes have ben corrected in the revised file of the manuscript.
I suggest to organize the manuscript further by adding subsections that should help the readers. For example, the section 4 “Testing the icosahedral…” could be divided by subheadings such as “Boron Neutron Capture Therapy”, “Metallabis(dicarbollides) cytotoxicity”, “SR-FTIRM studies”, “Contrast agents”, “Proton therapy, “antimicrobial activity”. Similarly, the authors could divide section 5 in subsections for clarity.
We have follow the referee suggestion. Thanks a lot for them.
Reviewer 2 Report
The authors systematically summarized their recent work on the new applications of boron icosahedral clusters as the key components in the field of novel healthcare materials, based on the stability-aromaticity relationship and on the progress made in the synthesis of derivatized clusters, including boranes, carboranes and metallabis(dicarbollides). They indicated that these 3D geometric shape clusters, the semi-metallic nature of boron, and the presence of exo-cluster hydrogen atoms could interact with biomolecules through non-covalent hydrogen or dihydrogen bonds, which play a key role in endowing these compounds with unique properties in largely unexplored (bio)materials. The manuscript is well organized and represents an important contribution to the inorganic chemistry, boron chemistry, medicinal chemistry, and biochemistry. As such, I think that this work will be of interest for the interdisciplinary readership of Molecules. and should be considered for publication without change.
Author Response
Review 2:
Comments and Suggestions for Authors
The authors systematically summarized their recent work on the new applications of boron icosahedral clusters as the key components in the field of novel healthcare materials, based on the stability-aromaticity relationship and on the progress made in the synthesis of derivatized clusters, including boranes, carboranes and metallabis(dicarbollides). They indicated that these 3D geometric shape clusters, the semi-metallic nature of boron, and the presence of exo-cluster hydrogen atoms could interact with biomolecules through non-covalent hydrogen or dihydrogen bonds, which play a key role in endowing these compounds with unique properties in largely unexplored (bio)materials. The manuscript is well organized and represents an important contribution to the inorganic chemistry, boron chemistry, medicinal chemistry, and biochemistry. As such, I think that this work will be of interest for the interdisciplinary readership of Molecules. and should be considered for publication without change.
Thanks so much to the referee for his/her report and comments that we appreciate very much.